# A Novel Hybrid Edge Detection and LBP Code-Based Robust Image Steganography Method

Habiba Sultana [1,*], A. H. M. Kamal [1], Gahangir Hossain [2] and Muhammad Ashad Kabir [3]

1   Department of Computer Science and Engineering, Jatiya Kabi Kazi Nazrul Islam University,
    Mymensingh 2220, Bangladesh; kamal@jkkniu.edu.bd
2   Department of Information Science, University of North Texas, Denton, TX 76203-5017, USA;
    gahangir.hossain@unt.edu
3   School of Computing, Mathematics and Engineering, Charles Sturt University, Bathurst, NSW 2795, Australia;
    akabir@csu.edu.au
*   Correspondence: srity.cse@gmail.com

**Abstract:** In digital image processing and steganography, images are often described using edges and local binary pattern (LBP) codes. By combining these two properties, a novel hybrid image steganography method of secret embedding is proposed in this paper. This method only employs edge pixels that influence how well the novel approach embeds data. To increase the quantity of computed edge pixels, several edge detectors are applied and hybridized using a logical OR operation. A morphological dilation procedure in the hybridized edge image is employed to this purpose. The least significant bits (LSB) and all LBP codes are calculated for edge pixels. Afterward, these LBP codes, LSBs, and secret bits using an exclusive-OR operation are merged. These resulting implanted bits are delivered to edge pixels' LSBs. The experimental results show that the suggested approach outperforms current strategies in terms of measuring perceptual transparency, such as peak signal-to-noise ratio (PSNR) and structural similarity index (SSI). The embedding capacity per tempered pixel in the proposed approach is also substantial. Its embedding guidelines protect the privacy of implanted data. The entropy, correlation coefficient, cosine similarity, and pixel difference histogram data show that our proposed method is more resistant to various types of cyber-attacks.

**Keywords:** steganography; LBP code; LSB; edge detector; PSNR; SSIM

## 1. Introduction

Modern communication is not complete without information security because it aids in preventing sensitive and confidential data from being accessed, altered, or destroyed by unauthorized parties. It guarantees that a person's reputation, financial resources, and privacy are all protected. Cryptography, information fusions, watermarking, and steganography are a few of the approaches used to maintain information security [1]. A message is encrypted using a key before being sent, concealing the original message inside a cipher. To obtain the original message, the receiver decrypts the cipher. There are a number of problems with cryptography [2]. By altering the contents of the message with a key, it encourages uncertainty in its meaning. Therefore, one can assume that the encrypted message might contain some sensitive information, which might motivate one to try to figure out how to decrypt the encrypted message [3]. Information from several sources is combined through information fusion. Additionally, they go through a complicated computing process [4]. Once more, the goal of watermarking is to guarantee data integrity. Additionally, it might show the mask that was implanted on the data [5,6].

Steganography, on the other hand, is the technique of concealing data within a cover media. It has the capacity to trick intrusions by acting as though the cover material is completely secret. Since steganography provides secure communication with privacy, it has numerous applications in areas where secrecy is crucial. It can be used in medical, military,

law enforcement, intelligence, and counterintelligence agencies. Commonly used cover media are images, audio, video, text, DNA sequences, etc. In the field of steganography, images are the most famous form of cover media for frequently communicating flexible, redundant content over the internet. The method in the state of the art is to implant secrets in either image pixels [7,8], transform coefficients of pixel values [9], or predict error space. In all cases, an embedding method, known as an encoder, implants the secrets within an image by modifying its pixel values. When the data embedment is complete, the media is termed a stego image. The stego image is then sent to the target destination. At the destination, a decoder extracts the secret from the stego image. Some decoders might be able to rebuild the cover media from the stego media without any help from the sender side. If the scheme can extract both the secret message and the cover, the method is called a reversible process. On the other hand, the irreversible schemes only extract the secrets [10–26]. Reconstruction of the cover image is not their concern. Though reversible processes hide fewer data, they are more challenging to implement [27–45]. However, in both cases, the performance of the schemes are measured by imperceptibility, security, capacity, robustness and embedding complexity, etc.

For enhanced data security, a very rigorous technique is to implant bits into edge pixels only. Such a scheme first applies an edge detection algorithm to find the edge pixels of an image. The edge pixels are then used to create data bits. Though these schemes have a low embedding capacity, they are famous for security reasons. Our goal is to hide information within those edge pixels. We studied several edge detection-based schemes. However, these are irreversible schemes. That is why our second goal is to make a reversible method that is similar to those ones.

Chen et al. [7] proposed a hybrid edge-based image steganography scheme. They combined canny and fuzzy logic-based edge images utilizing the OR operator in their method, and secret bits were implanted using the least significant substitution technique. Within image pixels, they kept track of the edge status. As a result, this design has a low embedding capacity as well as poor visual quality. To solve the problem of [7], Tseng and Leng [8] proposed block and hybrid edge-based image steganography methods. They also stored edge status information within image pixels, but that also suffers from poor visual quality. The scheme proposed in [15] overcame the drawbacks of [7,8]. In that scheme, authors utilize the full embedding space without storing edge information.

Sun [10] proposed a novel edge-based image steganography method. The Canny edge detection technique is used by the author to identify and implant only edge pixels in this scheme. They simultaneously used the $2^k$ correction approach to improve the image quality of stego images while encrypting the secret data using the Huffman encoding algorithm. The scheme suffers from poor embedding capacity due to the selection of edge pixels only. Swain [11] proposed another image steganography method based on pixel value differencing (PVD). This scheme suffers from poor embedding capacity. Khan et al. [12] proposed a true edge-based image steganography method. By utilizing a clever edge detection technique, they only choose edge pixels, and they use the LSB method to implant four bits of hidden information into each edge pixel. Along with having poor visual quality, this scheme also has weak embedding capability. Hussain et al. [13] proposed a new steganography scheme by combining irreversible and reversible methods. Their approach is based on pixel value difference (PVD), least significant bit (LSB) substitution, PVD shift, and change of prediction error (MPE). Al-Dmour and Al-Ani [14] proposed a novel image steganography method based on edge detection and XOR coding. They implant secrets either in the spatial domain or transform domain of the cover image.

The scheme of [3] combined cryptography and steganography algorithms. They encrypt the secret data using the Data Encryption Standard (DES) and implant using the LSB method on the edge area of the cover image, which is generated by the canny edge detection algorithm. Vanmathi and Prabu [16] also combined cryptography and steganography algorithms. They use the chaotic approach to encrypt the secret data, and they implant it using LSB and variable-length data that is hidden on the cover image's edge

and non-edge regions, which is produced by a fuzzy edge detector. The scheme of [17] proposed a novel image steganographic algorithm based on the canny edge detection algorithm and hybrid hamming codes. Gaurav and Ghanekar [18] proposed another steganography method based on canny edge detection, dilation morphological operator, and XOR coding. Kumar et al. [20] proposed an image steganography based on the fuzzy edge detection method. They detect edge pixels using the fuzzy edge detection algorithm on a 2-bits cleared image and implant 2–bits secrets on selected edge areas using the LSB method. Setiadi and Jumanto [19] proposed another image steganography method based on hybrid edge detection and LSB method. Setiadi [21] proposed an image steganography method based on the hybrid edge detection algorithm, the dilation morphological operator, and the LSB method.

Ghosal et al. [26] proposed an image steganography method based on the Kirsch edge detection algorithm and implant secrets into each triplet of pixels. The scheme of [22] proposed an image steganography method based on PVD and edge detection algorithm. Jan et al. [23] proposed an image steganographic algorithm based on a logistic map for encrypting secret information and Laplacian of Gaussian (LoG) edge operator to find edge areas of the cover image. Sultana and Kamal [25] proposed an image steganography scheme based on hybrid edge detection and LSB method. Lee et al. [39] proposed a reversible image steganography based on reduplicated exploiting modification direction, image interpolation, and canny edge detection algorithm. Kamal and Islam [32] proposed a multi-layer data embedment scheme based on prediction error. The scheme of [40–43] proposed a reversible image steganography algorithm. Chakraborty and Jalal [24] proposed an image steganography method based on local binary pattern (LBP). Kamal and Islam [27] proposed an image steganography algorithm based on prediction error and LBP code. Sahu et al. [44] proposed reversible image steganography based on the LBP code and the XOR operator.

This paper proposes a hybrid edge detection and LBP code-based image steganographic technique. The proposed scheme applies an *n*-number of edge detectors to the cover image, hybridizes those generated edge images using the OR operator, and also performs a dilation morphological operation. Only edge pixels are selected and arranged into a two-dimensional matrix to collect LBP code. Then the LPB code and secret bits are XORed and shuffled. Finally, the stego image is generated, and the changes to the cover image are tracked. Experimental results show that the proposed scheme performs better than the other competing methods. The key contributions of the paper are threefold.

- Our proposed scheme increases the number of edge pixels by hybridizing edge images with an OR operator and conducting further need-based dilatation of edge areas in the hybridized image, which in turn improves the embedding capacity.
- The scheme improves the data embedding capabilities and robustness of the technique by implanting data in generated LBP codes from edge pixels.
- The strategy also preserves the stego image's visual quality, which is higher than the competitors. The technique demonstrates considerable resistance to statistical assaults as well.

The rest of the article is organized into several sections. Section 2 provides the details of related works. The proposed method is presented in Section 3. Section 4 demonstrates the simulated results of our scheme and the testing results on the robustness of the proposed scheme against attacks. Finally, Section 5 ends the article.

## 2. Related Works

### 2.1. A Brief on Edge Detectors and LBP Code

An edge is a boundary between two distinct sections of an image, or it can be described as a group of contiguous pixel positions or abrupt changes in intensity values. Edges may exist vertically, horizontally, or diagonally. Edge detection is the process of segmenting an image into areas of discontinuity. The edge detectors Canny, Sobel, Log, Prewitt, Kirsch,

Laplacian, and Fuzzy are very widely utilized. In general, edge detectors are employed in digital image processing, pattern recognition, image morphology, feature extraction, etc.

A $3 \times 3$ kernel's center pixel and its eight surrounding pixels are compared to produce the local binary pattern or LBP. The visual properties of the image are represented by this 8-bit pattern, which adapts to non-uniform environmental changes. Consider a $3 \times 3$ pattern's block where $K_c$ and $K_p^i$ denote the center and other pixels of a block, respectively, and $1 \le i \le 8$. At the same time, we also take a sample image block of $3 \times 3$ pixels. The LBP method classifies the block contents into two regions-(i) pixel values greater than $k_c$, which are represented by 1, and (ii) the other pixels in the pattern, which are represented by 0. Figure 1 shows a step-by-step approach to generating LBP code values.

| $K_p^4$ | $K_p^3$ | $K_p^2$ |
|---|---|---|
| $K_p^5$ | $K_c$ | $K_p^1$ |
| $K_p^6$ | $K_p^7$ | $K_p^8$ |

| 23 | 47 | 65 |
|---|---|---|
| 91 | 83 | 12 |
| 96 | 64 | 34 |

**if** $K_p^i > K_c$ **then**
  $L^i \leftarrow 1$
**else**
  $L^i \leftarrow 0$
**end if**

| 1 | 1 | 1 |
|---|---|---|
| 0 | | 1 |
| 0 | 1 | 1 |

(**a**)  (**b**)  (**c**)  (**d**)

**Figure 1.** Generating LBP code: (**a**) the way of representing contents in a pattern; (**b**) a sample pattern; (**c**) the pseudo-code for LBP; (**d**) LBP code.

In order to keep the structural and statistical characteristics of an image while reducing the quantity of data in it, edge detection and LBP are used. Many publications on that topic have been studied by our team. Among those, we found the works of Chakraborty [24], Sultana [25], Sahu [44], and Kamal [27] and read them very carefully and attentively and built the foundation of our proposed work upon their findings.

### 2.2. LBP-Based Image Steganography Method

In 2020, Chakraborty et al. [24] proposed an LBP-based image steganography method. They take a cover image and divide it into $3 \times 3$ non-overlapping blocks. They collect LBP code from each block and XORed with secret message bits. Then, they shuffled those XORed values. At the same time, they also performed synchronization operations to preserve the local neighborhood relationship.

### 2.3. Hybrid Edge Detection Based Image Steganography Method

In 2021, Sultana et al. [25] proposed a hybrid edge detection-based image steganography method. They applied an n-number of edge detectors on m-bits cleared images and hybridizes those edge images using AND operator. Then they classified the cover pixels as edge and non-edge pixels. At the same time, they also encrypt the secret message using the chaotic method. They implant x-bits into edge pixels and y-bits into non-edge pixels, where x > y, and generate a stego image.

### 2.4. LBP-Based Reversible Image Steganography Method

In 2022, Sahu et al. [44] proposed the LBP-based reversible data-hiding technique. They partition the image into $3 \times 3$ non-overlapping blocks. They collect LBP codes from each block. They take eighteen embeddable bits and divide them into three segments where two segments contain 8 bits and the remaining is 2 bits. Then they XORed those three segments with the LBP code of one block and concatenate them. They again divide those XORed values into nine segments and each contains two consecutive bits. Next, they implant the first eight segments of bits by the LSB method in neighbor pixels of a block, and bits of segment nine are implanted into the center pixel of this block. In this way, all data are implanted. During data implantation, two stego images are generated. At the extraction phase, secret message bits and cover images are restored using the reverse process.

### 2.5. LBP in Prediction Error Based Image Steganography Method

In 2022, Kamal et al. [27] proposed LBP in the prediction error-based image steganography method. They modified the traditional LBP method. Before applying the LBP method, they pre-process the image and convert all pixels as odd or even. A pixel value goes to odd when it is less than its reference value otherwise it is even. They modified the LBP code as 1 and −1 and −1 is used in place of 0. Then, they compute the encoded error and encoded pixel values. They implant secret bits in the encoded error and generate stego errors. Finally, this scheme generates stego pixels by adding encoded pixel values with encoded errors.

### 3. Proposed Work

In this paper, edge-detection and LBP-based reversible data embedment schemes have been proposed to preserve the structural and statistical features among the pixels of an image. The proposed method hybridizes edge images using OR operator and selects only edge pixels. They arrange edge pixels in a two-dimensional matrix and generate LBP code from it. Then they XORed LBP code with message bits. The XORed values are then implanted in edge pixels and generate a stego image. Later, the original cover image pixels, as well as the message bits can be reversibly convertible from the produced stego image at the receiver end. Following this, the embedding, extraction, and cover image restoration steps are presented and this also graphically shown in Figure 2.

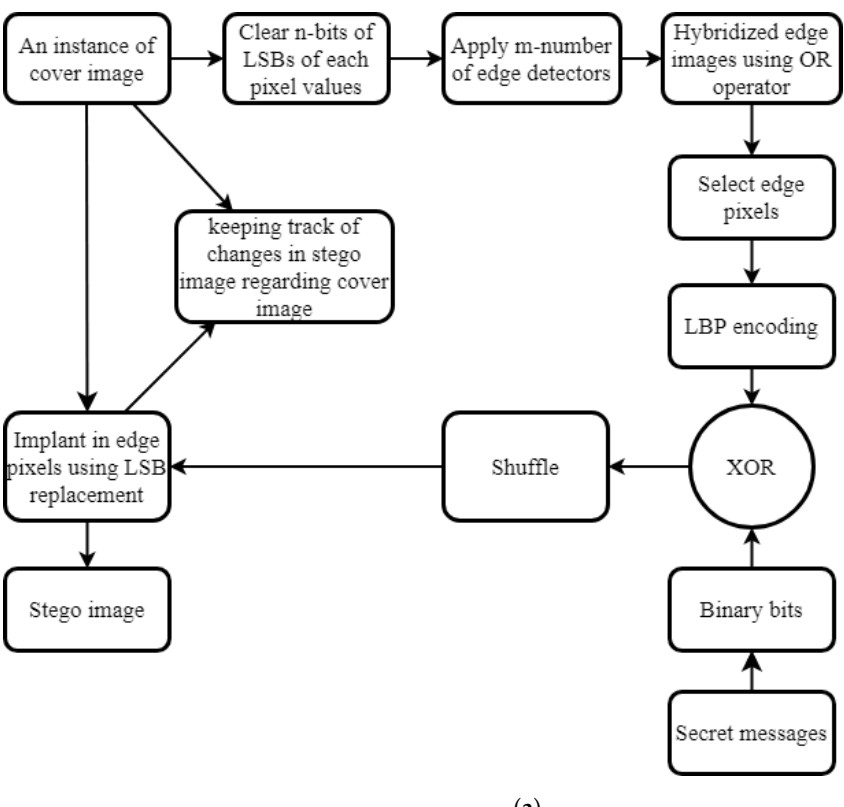

(**a**)

**Figure 2.** *Cont*.

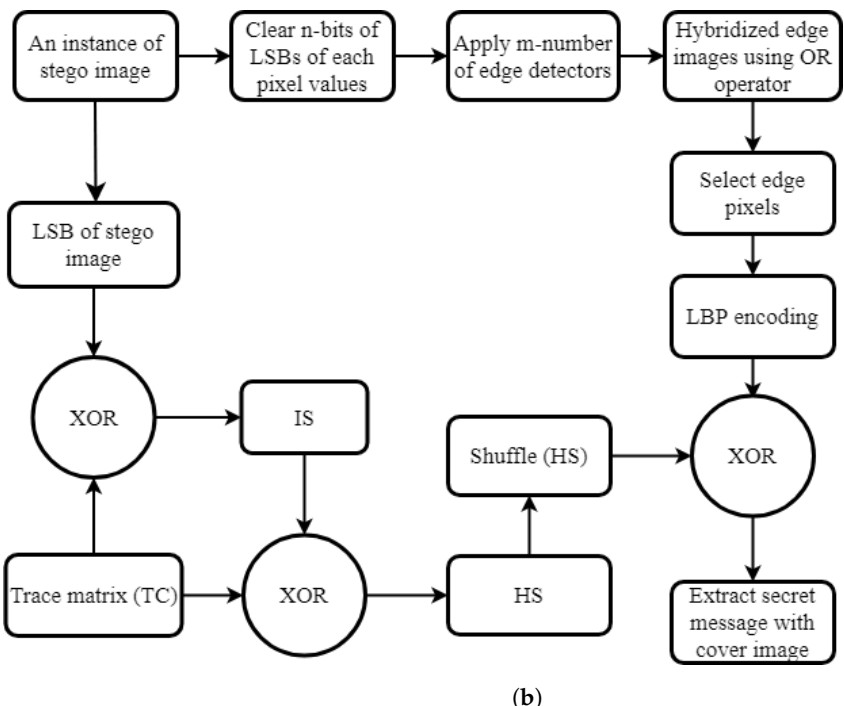

**(b)**

**Figure 2.** Proposed method scheme. (**a**) Embedding process; (**b**) Extraction process.

*3.1. Embedding Steps*

Step 1: Let the cover image be $C$ and an instance of it by $I$. We first clear $n-$bits of LSBs from every pixel of $I$ by Equation (1).

$$I(i,j) = I(i,j) - f(I(i,j), 2^n);  \qquad (1)$$

where function $f$ returns the remainder value when one divides $I(i,j)$ by $2^n$.

Step 2: A $m-$number of edge detection operators are applied, e.g., canny, sobel, fuzzy, Robert, Prewitt, log, etc., on the cleared image I to detect edge pixels, separately. The edge image is generated by Equation (2).

$$eI(i) = \psi(I, \Omega);  \qquad (2)$$

where $\psi$ is one of the $m$ edge detection operators, i.e., $\Omega \in \{canny, sobel, log, fuzzy, Robert, Prewitt, etc.\}$ and $1 <= I <= m$ and $\psi$ returns the edge image $eI$ from $I$ for a specific edge detector $\Omega$. All possible combinations of edge images can be made if it is required. The edge images are then hybridized using the logical OR operator by Equation (3). Each edge image is a binary image. For each pixel, the edge image holds a 0 or 1. A 1-in-edge image means the corresponding pixel of $I$ is in the detected edge.

$$ReI = eI(1)|eI(2)|.......|eI(m);  \qquad (3)$$

The hybridized edge image helps us in measuring the edge pixels in an image. To meet a bit higher embedding capacity, the number of edge pixels was increased by applying a morphological dilation operation by Equation (4). The emphasis was set on horizontal and vertical edges. Hence, the dilation operator, i.e., mask, is prepared by a $3 \times 3$ matrix.

$$dReI = F(ReI, mask);  \qquad (4)$$

where $F$ returns the dilated edge images and we then select only edge pixels. Say, the detected edge pixels have formed another image $P$. That image $P$ is partitioned into different $3 \times 3$-sized blocks.

Step 3: Let L be the 8-bit LBP codes that are obtained from the block using Equations (5) and (6).

$$g(K_c, K_p^i) = \begin{cases} 1 & \text{if } K_p^i < K_c \\ 0 & \text{else} \end{cases} \tag{5}$$

$$L = g(K_c, K_p^i); \tag{6}$$

Step 4: Now, LBP code $L$ was XORed with message bits $M$ by Equation (7).

$$H = L \oplus M; \tag{7}$$

Step 5: Next, the XORed values were shuffled by Equation (8).

$$HS = Shuffle(H); \tag{8}$$

Step 6: The shuffle bits are implanted in the edge pixels of the cover image and generate a stego image by following the way. At the same time, the changes in the stego block regarding cover image pixels were surveilled.

**if** $HS(i) = 1$ **then**
    **if** $C(i)$ is odd **then**
        $SB(i) \leftarrow C(i)$
    **else**
        **if** $C(i)$ is even **then**
            $SB(i) \leftarrow C(i) + b$
        **end if**
    **end if**
**else**
    **if** $HS(i) = 0$ **then**
        **if** $C(i)$ is even **then**
            $SB(i) \leftarrow C(i)$
        **else**
            **if** $C(i)$ is odd **then**
                $SB(i) \leftarrow C(i) + b$
            **end if**
        **end if**
    **end if**
**end if**

Step 7: Embedding is completed.

### 3.2. Data Extraction Cover Image Restoration

Step 1: Let the stego image is $SI$ and an instance of it by $S$. First, $n-$bits of LSBs are cleared from every pixel of $S$ by Equation (9).

$$S(i,j) = S(i,j) - f(S(i,j), 2^n); \tag{9}$$

where function $f$ returns the remainder value when one divides $S(i,j)$ by $2^n$.

Step 2: We have applied $m-$number of edge detection operators, e.g., canny, sobel, fuzzy, Robert, Prewitt, log, etc., on the cleared image I to detect edge pixels, separately. We have generated the edge image by Equation (10).

$$eS(i) = \psi(S, \Omega); \tag{10}$$

where $\psi$ is one of the $m$ edge detection operators, i.e., $\Omega \in \{canny, sobel, log, fuzzy, Robert, Prewitt, etc.\}$ and $1 <= I <= m$ and $\psi$ returns the edge image $eI$ from $I$ for a specific edge detector $\Omega$. We also make all possible combinations of edge images if needed. We then hybridize edge images using the logical OR operator by Equation (11). Each edge image is

a binary image. For each pixel, the edge image holds a 0 or 1. A 1-in-edge image means the corresponding pixel of *I* is in the detected edge.

$$ReS = eS(1)|eS(2)|.......|eS(m); \tag{11}$$

The hybridized edge image helps in measuring the edge pixels in an image. To meet a bit higher embedding capacity, the number of edge pixels has been increased by applying a morphological dilation operation by Equation (12). Emphasis was given on horizontal and vertical edges. Hence, the dilation operator, i.e., mask, is prepared by a $3 \times 3$ matrix.

$$dReS = F(ReS, mask); \tag{12}$$

where *F* returns the dilated edge images and we then select only edge pixels. Say, the detected edge pixels have formed another image *P*. That image *P* is partitioned into different $3 \times 3$-sized blocks.

Step 3: Let L be the 8-bit LBP codes that are obtained from the block using Equations (13) and (14).

$$g(SK_c, SK_p^i) = \begin{cases} 1 & \text{if } SK_p^i < SK_c \\ 0 & \text{else} \end{cases} \tag{13}$$

$$SL = g(SK_c, SK_p^i); \tag{14}$$

Step 4: Now trace matrix TC is XORed with LSB of Stego block SB by Equation (15).

$$IS = TC \oplus LSB(SB); \tag{15}$$

Step 5: Again, the trace matrix IS is XORed with trace matrix TC by Equation (16).

$$HS = IS \oplus TC; \tag{16}$$

Step 6: Next, the XORed values are shuffled by Equation (17).

$$H = Shuffle(HS); \tag{17}$$

Step 7: To extract the secret message, again shuffled bits H XORed with LBP code SL by Equation (18).

$$M = H \oplus SL; \tag{18}$$

Step 8: The original edge pixels of the cover image were recovered and a cover image is generated by following the way.

**if** $SL(i) = 1$ **then**
　　$C(i) \leftarrow S(i) + TC(i)$
**else**
　　$C(i) \leftarrow S(i) - TC(i)$
**end if**

Step 9: Data extraction and cover image restoration are completed.

*3.3. Illustration of the Proposed Work*

An illustration of the proposed work is presented here. For explanation, we take nine edge pixels of a block from a hybridized edge image and skip the first two steps. Let the nine edge pixels of a cover image be $K_1 = 23$, $K_2 = 47$, $K_3 = 65$, $K_4 = 91$, $K_c = 83$, $K_5 = 12$, $K_6 = 96$, $K_7 = 64$ and $K_8 = 34$. We obtain LBP codes using Equations (5) and (6) from the block as $L = \{1, 1, 1, 1, 0, 0, 1, 1\}$. Assume that the secret message bits $M = \{0, 1, 0, 0, 0, 0, 0, 1\}$. Now, using Equation (7), we get the XORed values H.
$H = L \oplus M$ ;

$H = \{1, 1, 1, 1, 0, 0, 1, 1\} \oplus \{0, 1, 0, 0, 0, 0, 0, 1\};$
$H = \{0, 1, 1, 1, 0, 0, 0, 1\};$

Next, we shuffle the XORed values.
$H = \{1, 0, 1, 1, 0, 0, 1, 0\};$
Now using step 7, we implant the shuffle bits in edge pixels and generate stego pixels. At the same time, we also keep the trace matrix TC of this block. After implantation, stego pixels are $SK_1 = 23$, $SK_2 = 47$, $SK_3 = 65$, $SK_4 = 92$, $SK_c = 83$, $SK_5 = 12$, $SK_6 = 96$, $SK_7 = 64$ and $SK_8 = 33$ and the trace matrix is $TC = \{0, 0, 0, 0, 1, 0, 0, 1\}$. That stego image and trace matrices are sent to the receiver end.

At the extraction phase, consider the stego block from the hybridized edge image and the stego pixels are $SK_1 = 23$, $SK_2 = 47$, $SK_3 = 65$, $SK_4 = 92$, $SK_c = 83$, $SK_5 = 12$, $SK_6 = 96$, $SK_7 = 64$ and $SK_8 = 33$. Using Equations (13) and (14), we collect LBP code $SL = \{1, 1, 1, 1, 0, 0, 1, 1\}$ and trace matrix is $TC = \{0, 0, 0, 0, 1, 0, 0, 1\}$. Now, we XORed the trace matrix TC with the LSB of this block SB.
$IS = TC \oplus LSB(SB);$
$IS = \{0, 0, 0, 0, 1, 0, 0, 1\} \oplus \{0, 1, 1, 1, 0, 0, 0, 1\};$
$IS = \{0, 1, 1, 1, 1, 0, 0, 0\};$

Again, we XORed that value Is with trace matrix TC.
$HS = IS \oplus TC;$
$HS = \{0, 1, 1, 1, 1, 0, 0, 0\} \oplus \{0, 0, 0, 0, 1, 0, 0, 1\};$
$HS = \{0, 1, 1, 1, 0, 0, 0, 1\};$

Now, we shuffle the XORed values of HS.
$H' = \{1, 0, 1, 1, 0, 0, 1, 0\};$

To extract the secret message bits, we again XORed $H'$ with LBP code SL.
$M' = H' \oplus SL;$
$M' = \{1, 0, 1, 1, 0, 0, 1, 0\} \oplus \{1, 1, 1, 1, 0, 0, 1, 1\};$
$M' = \{0, 1, 0, 0, 0, 0, 0, 1\};$

Those are our implanted secret bits which are extracted successfully.

Using step 8 of the data extraction process, we are able to restore the original cover image pixels and those are $K_1 = 23$, $K_2 = 47$, $K_3 = 65$, $K_4 = 91$, $K_c = 83$, $K_5 = 12$, $K_6 = 96$, $K_7 = 64$ and $K_8 = 34$.

The message bits can be successfully extracted and the cover image can be restored. Thus, the proposed work is reversible.

## 4. Results and Discussion

This section shows the experimental results conducted to evaluate the performance of the proposed scheme compared with the works of Chakraborty [24], Sultana [25], Sahu [44], and Kamal [27]. First, ten frequently used images and an image dataset were selected. The experiment is set up and then the results are analyzed.

### 4.1. Experimental Setup

The experiments were performed on a desktop that is specified by an Intel (R) Core (TM) i5-8500T CPU @ 2.10 GHz 2.11 GHz processor and RAM of 8.00 GB. MATLAB R(2017a) was used on windows 7. In the proposed system, we used two separate forms of input data: the cover image and the secret message, which are both intended to be implanted data. Then, as indicated in Figure 3, we obtained a few representative texts from various sources. A text, binary, or other format for the sample message is possible. To convert the non-binary input data to binary, we used our prepared function ConBin. As an illustration, the ASCII values of text data are translated into binary. We deal with a range of message lengths. We are

unable to display all messages due to space complexity. As a cover media, ten frequently used standard images were collected as shown in Figure 4 to conduct all primary experiments. The corresponding stego images are shown in Figure 5. In addition, The BOSS dataset's 499 images were also analyzed. We changed the images' color to grayscale and scaled them to 512 × 512. We used pixel intensities because the dataset's contents were images.. The performance of the algorithm was measured with several feature values, such as edge pixel generation capability, embedding capacity, peak signal-to-noise ratio (PSNR), structural similarity index matrix (SSIM), correlation coefficient, entropy, cosine similarity, and Pixel difference histogram, etc.

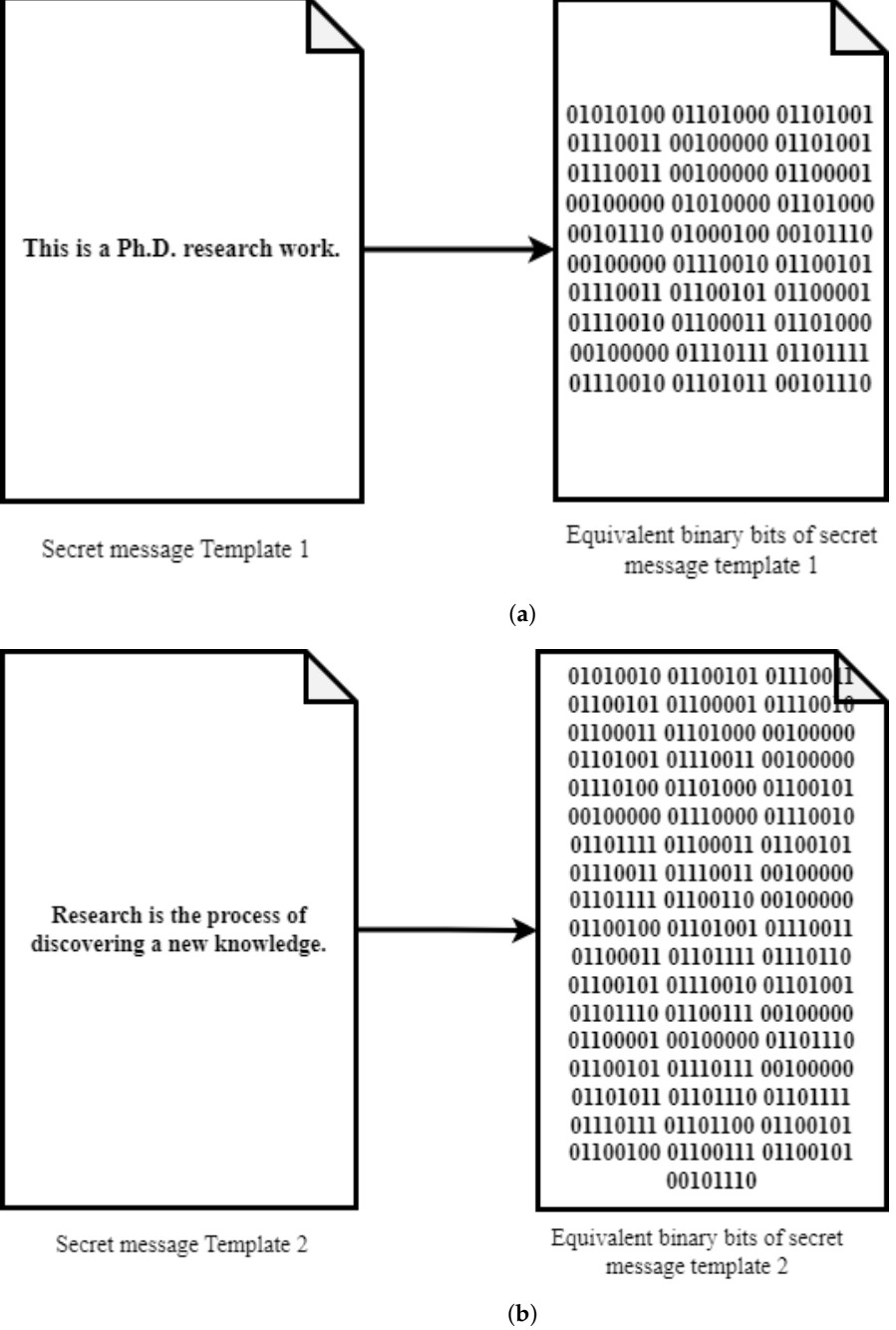

**Figure 3.** Secret message templates for the experiment (**a**) Secret message templates for the experiment; (**b**) Secret message templates for the experiment.

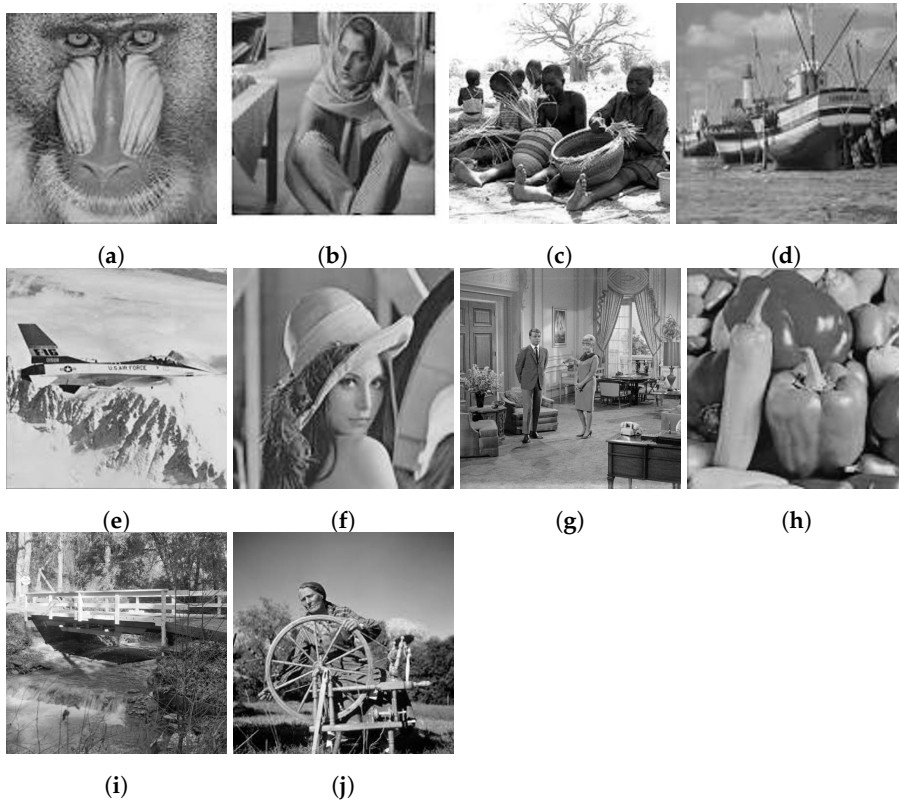

**Figure 4.** Cover images for the experiment (**a**) Baboon, (**b**) Barbara, (**c**) Basket, (**d**) Boat, (**e**) F16, (**f**) Lena, (**g**) Livingroom, (**h**) Peppers, (**i**) Walkbridge, (**j**) Wheel.

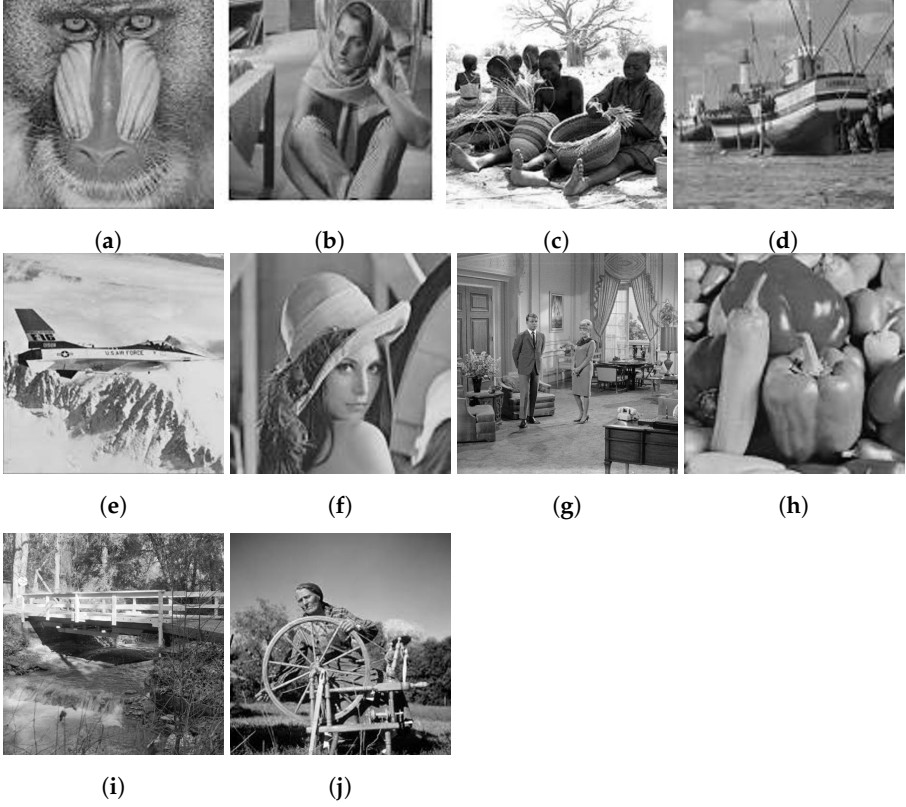

**Figure 5.** Stego images for the corresponding cover images: (**a**) Baboon, (**b**) Barbara, (**c**) Basket, (**d**) Boat, (**e**) F16, (**f**) Lena, (**g**) Livingroom, (**h**) Peppers, (**i**) Walkbridge, (**j**) Wheel.

*4.2. Mathematical Representation of Feature Values*

The embedding capacity was measured, as the number of implanted bits per tempered pixel, which is shown as Embedding capacity *EC* by Equation (19)

$$EC = \frac{P}{T_p};$$ (19)

where, *P* is the total number of implanted bits in the cover image, and $T_p$ is the number of tempered pixels. Tempered pixels contain conceive bits or associate rules of embedding.

Maintaining image quality is a challenging task and for this purpose, peak-signal-to-noise-ratio (PSNR) and structural similarity index measure (SSIM) are commonly used as image distortion measurement parameters. The PSNR is measured by Equation (20)

$$PSNR = 10 \log_{10} \frac{255^2}{MSE};$$ (20)

where,

$$MSE = \frac{1}{h \times w} \sum_{i=1}^{w} \sum_{j=1}^{h} (S_{i,j} - C_{i,j})^2;$$ (21)

where, *S* is the stego image and *C* is the original cover image. Next, the SSIM is calculated by Equation (22)

$$SSIM = \frac{(2\mu_c\mu_s + C_1)(2\sigma_{cs} + C_2)}{(\mu_c^2 + \mu_s^2 + C_1)(\sigma_c^2 + \sigma_s^2 + C_2)}$$ (22)

where, $\mu_c$ and $\sigma_c$ are the mean and variance of pixel values in the cover image. Likewise, cover $\mu_s$, and $\sigma_s$ are the same for the stego image. $C_1$ and $C_2$ are two constants and we set $C_1 = 0.0001$ and $C_2 = 0.0009$ for experiment.

There are many methods of analyzing the robustness against various attacks. Famous techniques are entropy measurement, analyzing correlation among the pixels, checking the cosine similarity between the cover and stego image, and histogram of the Pixel difference between the stego and cover image. The entropy is measured by Equation (23)

$$H = -\sum_k P_k \log_2(P_k);$$ (23)

where, $P_k$ is the probability associated with gray value *k* and $1 \leq k \leq 255$.

Population correlation is defined by Equation (24)

$$P_{cs} = \frac{\sigma_{cs}}{\sigma_c\sigma_s};$$ (24)

where $\sigma_c$ and $\sigma_s$ are population correlation in cover C and stego S. Again, $\sigma_{cs}$ is the co-variance between the cover and stego image.

Equation (25) calculates the cosine similarity values

$$f_{\cos sim}(C, S) = \cos\theta = \frac{\sum_{i=1}^{h} \sum_{j=1}^{w} C(i,j) S(i,j)}{\sqrt{\sum_{i=1}^{h} \sum_{j=1}^{w} C(i,j)} \sqrt{\sum_{i=1}^{h} \sum_{j=1}^{w} S(i,j)}};$$ (25)

where *C* and *S* are cover and stego images.

*4.3. Experimental Results and Discussion*

In the experiment, Canny, Sobel, and Log, edge detectors are applied in five LSB-cleared images to identify edge and non-edge pixels. Canny-, Sobel-, and Log-based edge detector functions of MATLAB return an edge image for a given input image. The resultant edge image is a binary image. Next, those edge images are hybridized using the logical OR

operator. To increase the number of edge pixels, a morphological dilation operation was also performed. Only edge pixels are selected from the cover image using a hybridized image; it is an edge image. The edge image is divided into different $3 \times 3$ block sizes and generates LBP code for specific blocks and implant data according to embedding rules. The payload also calculated and analyzed the performance in embedding capacity with respect to tempered pixels. Embedding capacity is graphically shown in Figure 6. Figure 6 shows that our proposed scheme has the lowest embedding capacity with respect to Sahu's [44] and Sultana's [25] since secret messages are implanted in edge pixels only, and others are embedded in all the pixel values of images. Considering edge pixels, the proposed method demonstrates dominant performance.

The visual quality and structural originality of stego images are also analyzed. Visual quality is measured by PSNR values and is sketched in Figure 7. It is clear from the diagram that the proposed scheme has a higher PSNR value than the competing schemes. The structural similarity index value, SSIM is also shown in Figures 8 and 9. Both figures show that the proposed scheme has the highest SSIM values than other schemes.

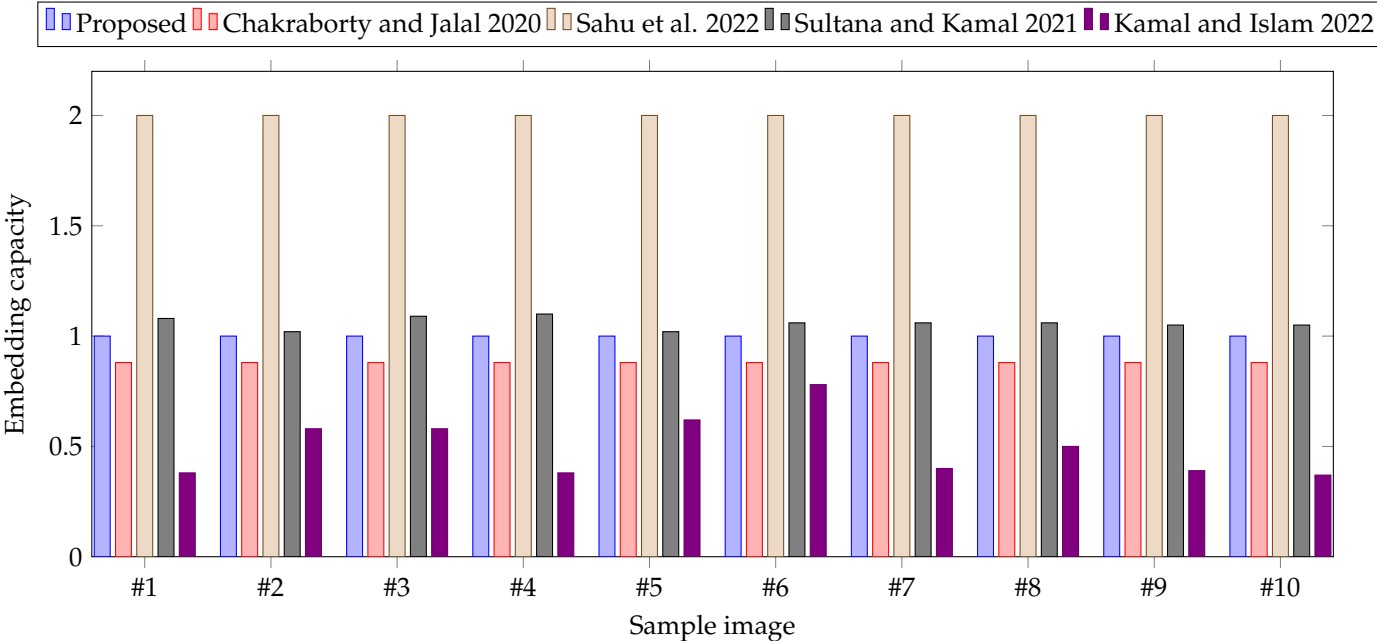

**Figure 6.** Embedding capacities of different images with respect to tempered pixels which are obtained by different schemes. The figure states that the proposed scheme has the third lowest embedding capacity due to implanting only edge pixels [24,25,27,44].

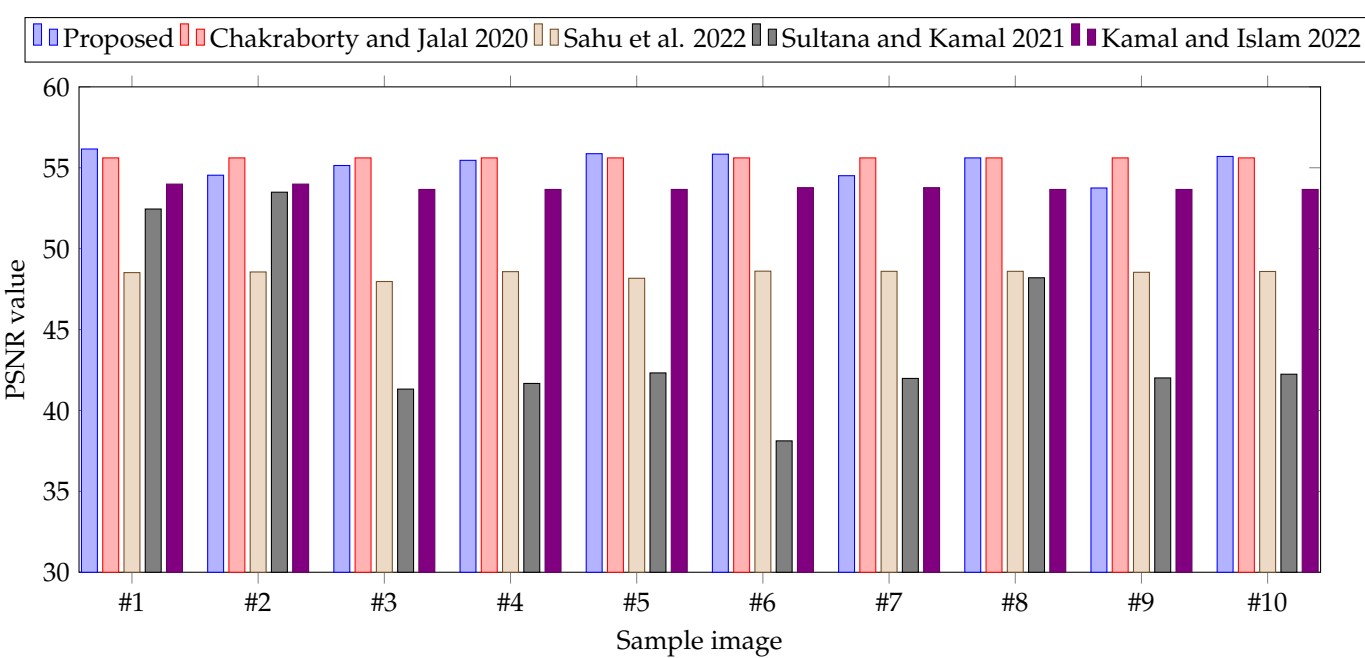

**Figure 7.** PSNR of different images in different schemes. The figure states that the proposed scheme has the highest PSNR of maximum images [24,25,27,44].

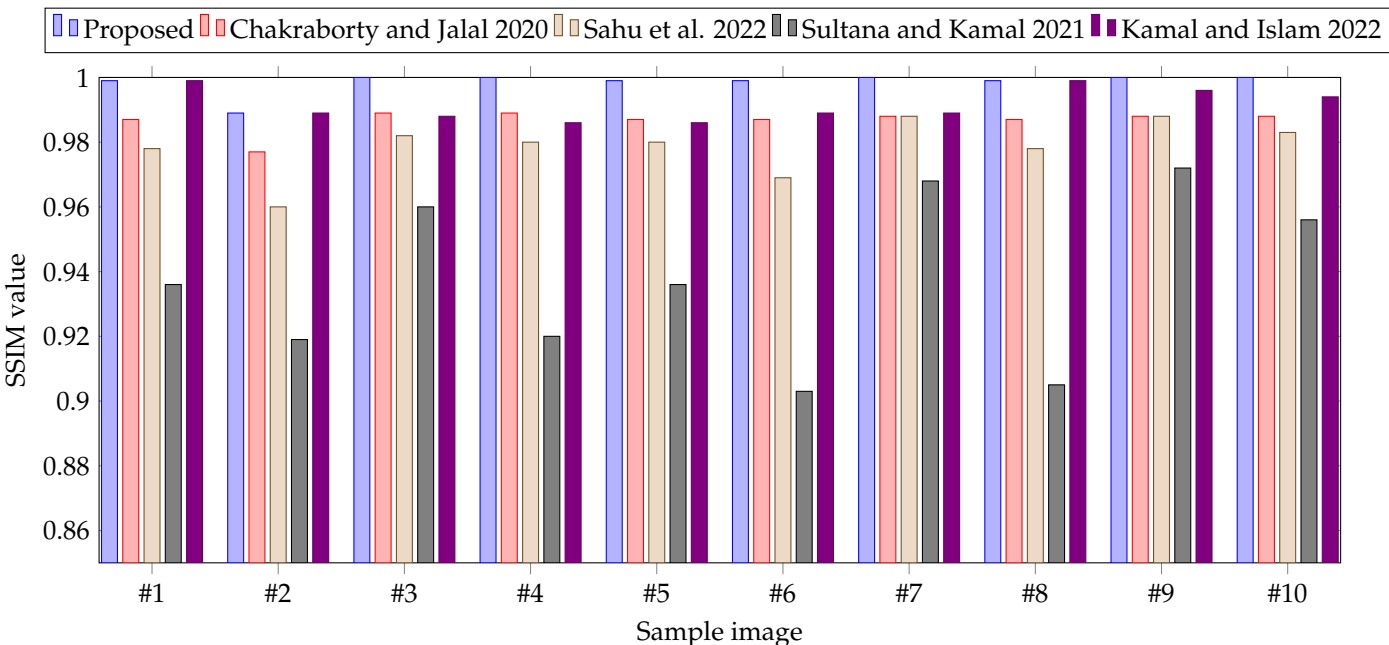

**Figure 8.** Comparing SSIM of different images in different schemes. The figure states that the proposed scheme has the highest SSIM value [24,25,27,44].

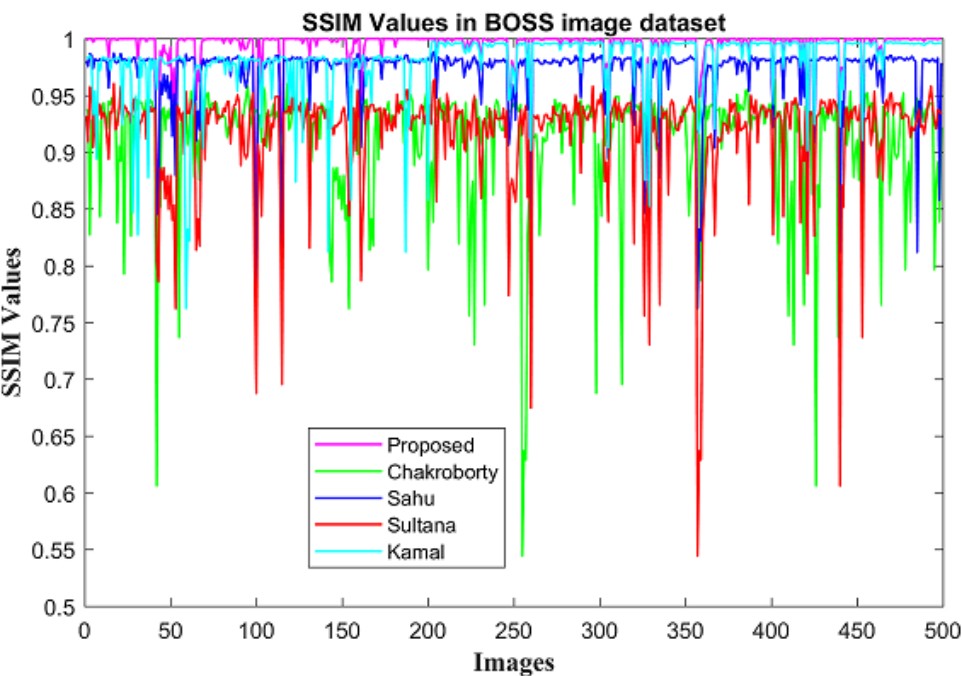

**Figure 9.** Comparing SSIM values of BOSS image dataset in different schemes. The figure states that the proposed scheme has the highest SSIM value.

### 4.4. Statistical Analysis

The novel approach was statistically analyzed using various parameters such as correlation coefficient, entropy, cosine similarities, and pixel difference histogram to check its robustness against various attacks. First, the correlation coefficients $\rho_s C$ were measured between the cover and stego image. $\rho_s C = 0$ stands for no relationship between two images. $\rho_s C > 0$ means a positive correlation between the cover and stego image and lies at a perfect relationship when it reaches 1. Similarly, a negative value of $\rho_s C$ indicates a negative relationship. Results of $\rho_s C$ are depicted in Table 1. Though the proposed method shows a higher correlation value, its difference from others is insignificant. Rather, as with others, it represents a higher correlation between the cover and the stego image.

**Table 1.** Correlation coefficient values of various schemes.

| Image Name | Correlation Coefficient Values | | | | |
|---|---|---|---|---|---|
| | Proposed | Chakroborty [24] | Sahu [44] | Sultana [25] | Kamal [27] |
| F16.jpg | 0.9999 | 0.9998 | 0.9993 | 0.9973 | 0.9998 |
| babon.jpg | 0.9999 | 0.9997 | 0.9990 | 0.9960 | 0.9997 |
| basket.jpg | 0.9999 | 0.9999 | 0.9998 | 0.9991 | 0.9997 |
| boat.jpg | 0.9999 | 0.9998 | 0.9993 | 0.9967 | 0.9999 |
| brbra.jpg | 0.9999 | 0.9999 | 0.9997 | 0.9988 | 0.9990 |
| lena.jpg | 0.9999 | 0.9998 | 0.9994 | 0.9971 | 0.9999 |
| livingroom.jpg | 0.9999 | 0.9998 | 0.9993 | 0.9970 | 0.9998 |
| pepper.jpg | 0.9999 | 0.9998 | 0.9995 | 0.9976 | 0.9998 |
| walkbridge.jpg | 0.9999 | 0.9999 | 0.9995 | 0.9980 | 0.9984 |
| wheel.jpg | 0.9999 | 0.9998 | 0.9996 | 0.9982 | 0.9996 |

The entropy values $H$ were also computed in both cover and stego images. Next, their difference was calculated. That difference value is zero for two identical images. Results are plotted in Figure 10. The figure shows that none of the results are greater than 0.06, i.e., these are very small and close to zero.

To verify further with similar statistics, cosine similarities are measured between the cover and stego images. That value is 1 for two identical images and 0 for two fully mismatched images. The results are demonstrated in Table 2. That table illustrates that our proposed method shows higher values than the other competing schemes.

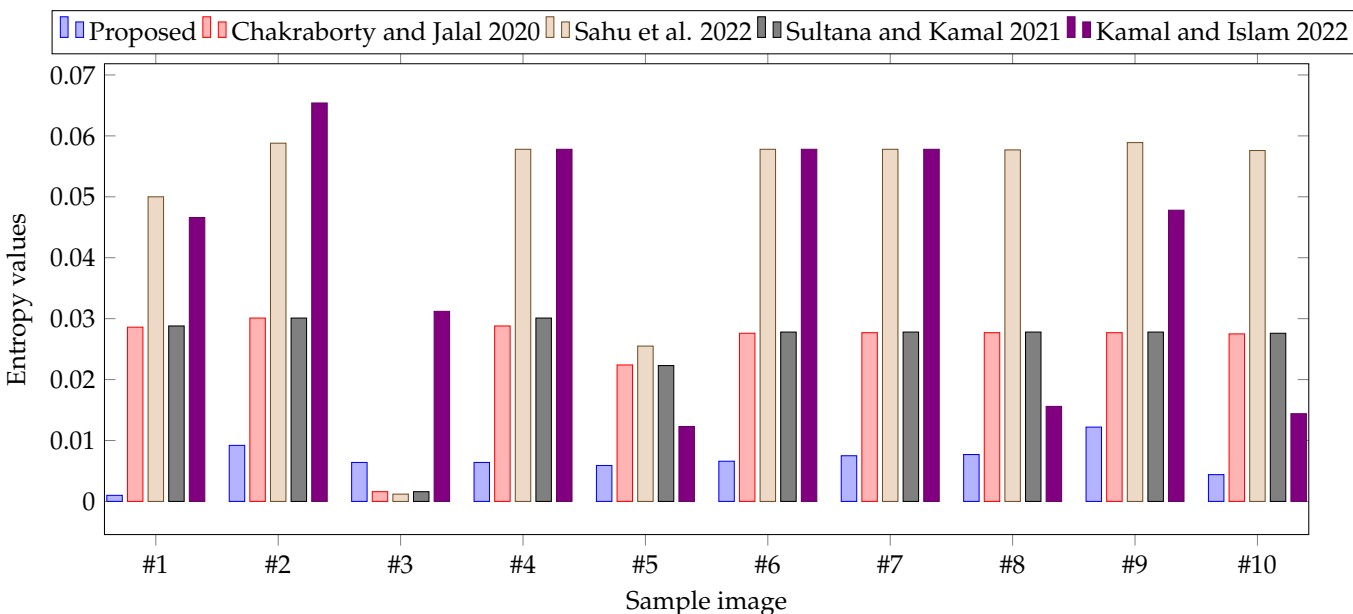

**Figure 10.** Performance comparison of the proposed scheme with the competing scheme in terms of entropy values. The figure states that the proposed scheme has the lowest entropy values [24,25,27,44].

**Table 2.** Cosine Similarity values of various schemes.

| Image Name | Cosine Similarity Values | | | | |
|---|---|---|---|---|---|
| | Proposed | Chakroborty [24] | Sahu [44] | Sultana [25] | Kamal [27] |
| F16.jpg | 0.99999 | 0.99999 | 0.99996 | 0.99984 | 0.99999 |
| babon.jpg | 0.99985 | 0.99985 | 0.99980 | 0.99960 | 0.99986 |
| basket.jpg | 0.99999 | 0.99991 | 0.99995 | 0.99981 | 0.99997 |
| boat.jpg | 0.99999 | 0.99998 | 0.99993 | 0.99966 | 0.99999 |
| brbra.jpg | 0.99999 | 0.99998 | 0.99994 | 0.99977 | 0.99990 |
| lena.jpg | 0.99937 | 0.99939 | 0.99934 | 0.99939 | 0.99939 |
| livingroom.jpg | 0.99999 | 0.99998 | 0.99992 | 0.99963 | 0.99998 |
| pepper.jpg | 0.99999 | 0.99998 | 0.99992 | 0.9996 | 0.99999 |
| walkbridge.jpg | 0.99999 | 0.99998 | 0.99992 | 0.99963 | 0.99984 |
| wheel.jpg | 0.99999 | 0.99998 | 0.99992 | 0.99966 | 0.99996 |

Aggregated result of big data set (BOSS dataset) is shown in Table 3. This table shows that our proposed scheme demonstrates strong resistance to attack from intruders.

**Table 3.** Aggregated values of statistical features of various schemes.

| Feature | Values | | | | |
|---|---|---|---|---|---|
| | Proposed | Chakroborty [24] | Sahu [44] | Sultana [25] | Kamal [27] |
| Entropy | 0.0032 | 0.0077 | 0.9973 | 1.4735 | 0.4783 |
| Standard Deviation | 3.4406 | 4.9651 | 5.9980 | 4.9960 | 3.9998 |
| Correlation coefficient | 0.9999 | 0.9969 | 0.9973 | 0.9981 | 0.9979 |
| Cosine Similarity | 0.9989 | 0.9893 | 0.9979 | 0.9966 | 0.9977 |

To detect the stego image, the pixel difference histogram (PDH), another statistical method is also used. Figures 11 and 12 show the PDH of the original images and associated

stego images, respectively. As a result, it can be inferred from the outcomes of these experiments that the proposed technique is robust enough to defend against attacks on implanted data.

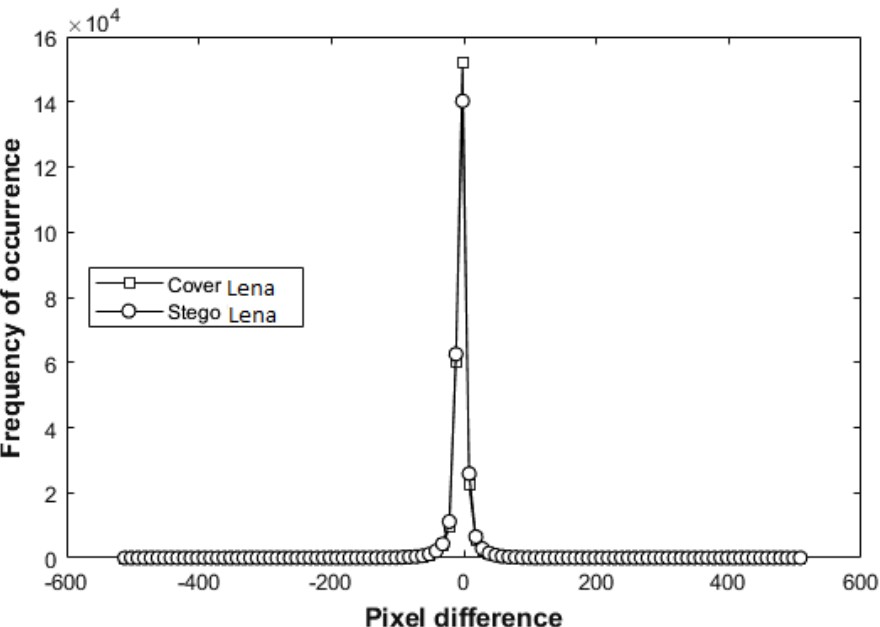

**Figure 11.** PDH plots of Lena for the proposed technique.

Histogram analysis is also another method to detect the difference between stego and cover image. Histograms are frequently used in statistics to show the frequency of a particular type of variable within a given range. Figures 13 and 14 show the histogram analysis of the original images and associated stego images, respectively. As a result, it can be inferred from the outcomes of these experiments that the differences between cover and stego are small and the proposed technique is robust enough to defend against attacks on implanted data.

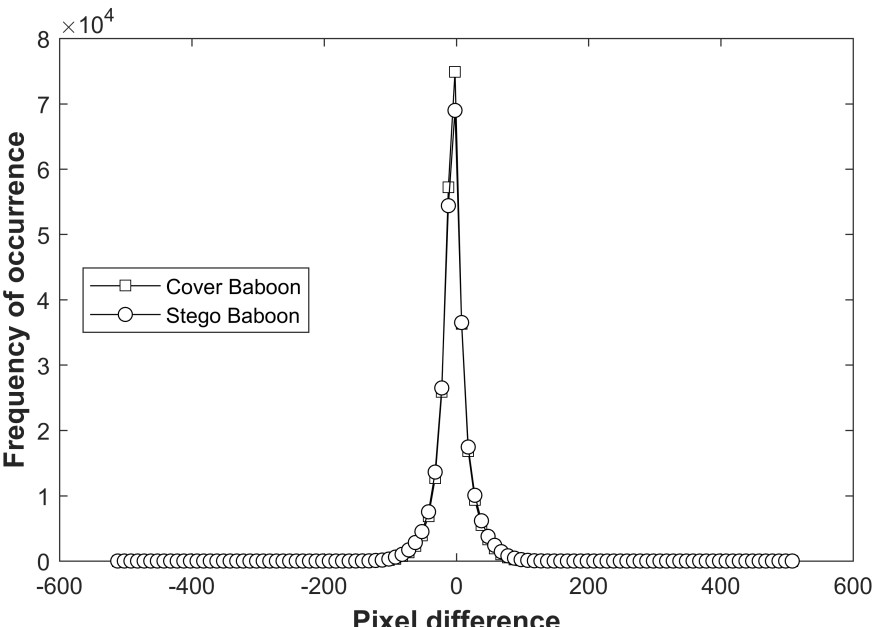

**Figure 12.** PDH plots of Baboon for the proposed technique.

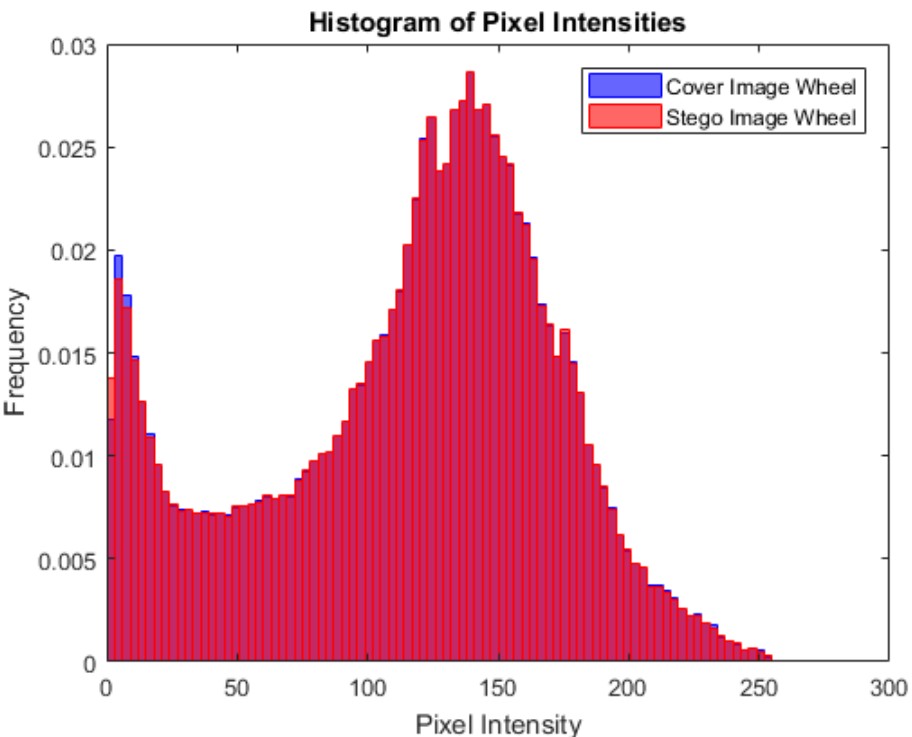

**Figure 13.** Histogram analysis plots of Wheel for the proposed technique.

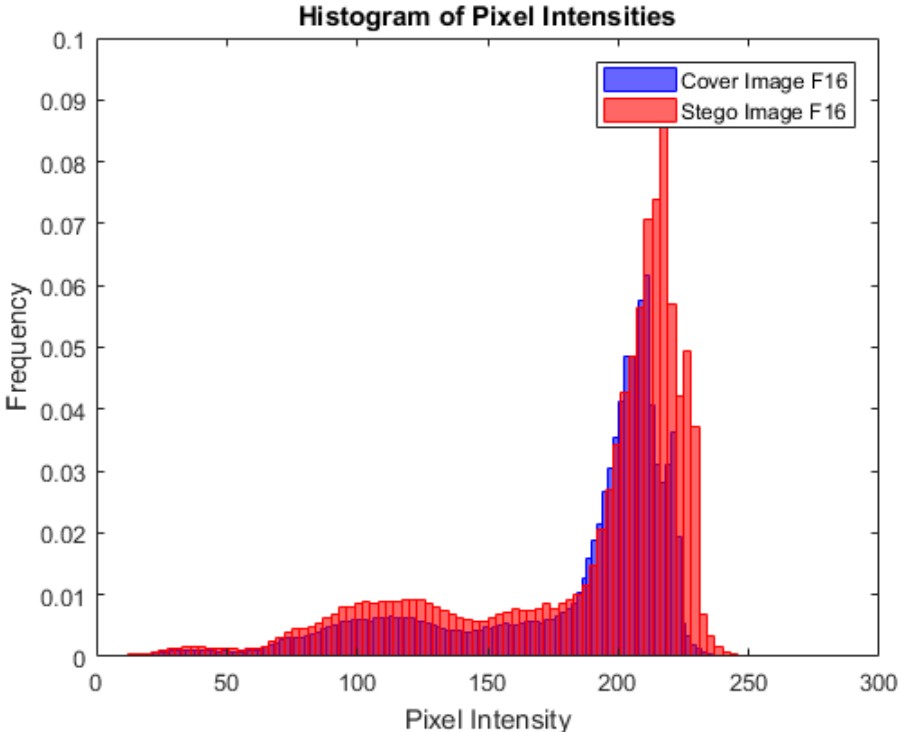

**Figure 14.** Histogram analysis plots of Wheel for the proposed technique.

$t$-test is another important statistical tool that is used to perform a hypothesis on one or a pair of datasets. It, indeed, has the capability to assess whether there is a significant difference between the means of two populations. That feature inspires us to use it in our statistical analysis to detect the tolerance level of changes by our stego image. Here, we have compared the mean pixel intensities of a cover image and a stego image using a two-sample $t$-test. The equation of $t$-test is given in Equation (26)

$$t = \frac{\frac{\Sigma(D)}{N}}{\sqrt{\frac{\Sigma D^2 - \frac{\Sigma(D)^2}{N}}{N-1}}}; \qquad (26)$$

Here $D$ is a matrix of differences of pixel values of cover and stego images, $N$ means degree of freedom. A high $t$-value shows a large discrepancy between the two sets of pixel values, pointing to the possibility that the suspicious image contains secret data. The result of $t$-test is shown in Table 4. This table shows that, in most cases, the scheme generates a $t$ value that is smaller than 1.646. From the T-distribution table, we can infer that these images accept the null hypothesis at 5% confidence level. On the contrary, 40% of images reject the null hypothesis at 5% confidence level.

**Table 4.** $t$-Test values of proposed method.

| ImageName | $t$-Test Values |
| --- | --- |
| F16.jpg | 6.3822 |
| baboon.jpg | 3.0015 |
| basket.jpg | 0.3493 |
| boat.jpg | 0.0039 |
| brbra.jpg | 0.0391 |
| lena.jpg | 7.5305 |
| livingroom.jpg | 0.2402 |
| pepper.jpg | 0.0039 |
| walkbridge.jpg | 0.0088 |
| wheel.jpg | 2.1080 |

## 5. Conclusions

Steganography is the art of sending digital images securely from a sender to a recipient. This paper suggests a novel hybrid edge detection and LBP code-based image steganographic technique that is robust and useful in image data security and transmission. In this proposed approach, the $m$-numbers of edge detection operators are applied on $n$-bits cleared cover images. Then, those edge images were hybridized using the logical OR operator and morphological dilation operation. Only the edge pixels are selected, from the edge image. It was divided into $3 \times 3$ block sizes, and LBP codes were generated. Secret bits are implanted in a cover media and generated in the stego media. At the recipient's end, both the secret bits and cover media are extracted. Only edge pixels are selected and arranged into a two-dimensional matrix to collect LBP code, before the LPB code and secret bits are XORed and shuffled. Subsequently, the stego image is generated, and the changes to the cover image are tracked. Experimental results show that the proposed scheme performs better than the other competing methods. The proposed scheme demonstrates 27.45%, 36.87%, 60.21%, and 45.56% higher PSNR values than Chakroborty [24], Kamal [27], Sultana [25], and Sahu [44], respectively.

This suggested hybrid LBP code-based image steganographic strategy, combining edge detection and LBP code in image steganography opens a new era of research and applications. This universal embedding technique applies to all current-generation image steganography techniques, considerably enhancing their security performance. Additionally, this research combined edge detection and LBP code; however, the combination can be conducted in various ways to achieve the expected performance. The proposed method is reversible, so the embedding capacity is small. This paper investigated several statistical methods, including the entropy, correlation coefficient, cosine similarity, and pixel difference histogram, and the results show that the proposed method is more resistant to different types of cyber-attacks. Moreover, future work will emphasize the full utilization of the embedding space and increase the embedding capacity.

**Author Contributions:** Conceptualization, H.S. and A.H.M.K.; methodology, H.S. and A.H.M.K.; Experiment, H.S.; writing—original draft preparation, H.S.; writing—review and editing, A.H.M.K., G.H. and M.A.K. This work is a contribution of Ph.D. research work of H.S. All authors have read and agreed to the published version of the manuscript.

**Funding:** This research work is partially supported by the University Grants Commission of Bangladesh through its research project and the Information and Communication Technology division of the Ministry of Post, Telecommunication, and Information Technology of the Government of Bangladesh.

**Institutional Review Board Statement:** Not applicable.

**Informed Consent Statement:** Not applicable.

**Data Availability Statement:** We used data from "The Bank of Standardized Stimuli (BOSS), a New Set of 480 Normative Photos of Objects to Be Used as Visual Stimuli in Cognitive Research, Mathieu B. Brodeur, Emmanuelle Dionne-Dostie, Tina Montreuil, Martin Lepage" and various reliable sources (Internet).

**Acknowledgments:** Sultana is a Ph.D. fellow of the Information and Communication Technology division of the Ministry of Post, Telecommunication and Information Technology of the Government of Bangladesh. So, we would like to acknowledge their support.

**Conflicts of Interest:** The authors declare no conflict of interest.

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
