# Peer review of "A Novel Hybrid Edge Detection and LBP Code-Based Robust Image Steganography Method"

_futureinternet, doi:10.3390/fi15030108_

Round 1

Reviewer 1 Report

The performance evaluation section should presend a larger set of experiments, and a deeper statistical analysis. In particular, the experiments should be performed on a range of plaintexts and embedding images, and the performance should be presented under variation of parameters, reporting confidence intervals. Open science can be fostered by releasing source code and datasets for the fellow researchers to build upon and before acceptance for reviewers to independently verify the claims.

Reviewer 2 Report

This manuscript uses the morphological expansion procedure in the mixed-edge image. The least significant bit (LSB) of the edge pixels and all local binary pattern (LBP) codes are calculated. Afterward, the LBP codes, LSBs and secret bits are merged using heterogeneous operations. The experimental results show that the proposed method is superior to the current strategy in measuring perceptual transparency such as peak signal-to-noise ratio (PSNR) and structural similarity index (SSI). However, major revisions are needed before it is finally accepted.

1.     Its embedding guidelines protect the privacy of implanted data. The entropy, correlation coefficient, cosine similarity, and pixel difference histogram data show that the proposed method is more resistant to various cyber-attacks. However, it will be better if the authors can add some more simulation and experimental results. For example, the robustness of the proposed scheme is testable, authors can test the robustness by incorporating noise addition, random cropping, resampling, re-quantization, shifting, and matrix decomposition.

2.     The embedding capacity per tempered pixel in the proposed approach is also substantial. However, I found in the paper that the embedding capacity of Sahu's method is better than the method proposed in this study (as shown in Figure 3), and Sahu's method is a reversible information-hiding technique, does this study have other advantages over Sahu's method?

Reviewer 3 Report

In digital image processing and steganography, images are often described using edges and local binary pattern (LBP) codes. By combining these two properties, a novel hybrid image steganography method of secret embedding is proposed in this paper. This method only employs edge pixels that influence how well the novel approach embeds data and several edge detectors are applied and hybridized using a logical OR operation. Paper is well written, relevant and concise.

Round 2

Reviewer 1 Report

The authors have modified the manuscript following some of the reviewer's comments. However, a few issues remain. Namely, the limited statistical analysis of the results. A positive aspect is the use of a larger dataset. Aggregated performance metrics should be provided instead of image-by-image performance.

Reviewer 2 Report

The authors revised their manuscript according to the suggestions in my report. I recommend it for publication.

Round 3

Reviewer 1 Report

The authors have modified the manuscript following the reviewers' suggestions. There are elements of statistical analysis, such as the t-test, which go in the right direction. Before being considered for acceptance, the authors should fix a few minor points. In general, one can never "accept" the null hypothesis, but only fail to reject. Furthermore, the discussion about the t-test is minimal: only one sentence. Please extend it considerably to describe the meaning of the shown numbers. Space limitation is not an issue, as several figures are currently huge and can be shrunk.

Round 4

Reviewer 1 Report

The reviewers have extended the description of the statistical analysis used. As a minor last comment before considering the article for publication: the authors should consider modifying Figures 5, 6, 7, and 9 from line plots to bar charts, as there is not an ordered relation between the images.
